# Effects of UV and UV-vis Irradiation on the Production of Microalgae and Macroalgae: New Alternatives to Produce Photobioprotectors and Biomedical Compounds

**DOI:** 10.3390/molecules27165334

**Published:** 2022-08-22

**Authors:** Rafael G. Araújo, Brian Alcantar-Rivera, Edgar Ricardo Meléndez-Sánchez, María Adriana Martínez-Prado, Juan Eduardo Sosa-Hernández, Hafiz M. N. Iqbal, Roberto Parra-Saldivar, Manuel Martínez-Ruiz

**Affiliations:** 1Tecnológico de Monterrey, Institute of Advanced Materials for Sustainable Manufacturing, Monterrey 64849, Mexico; 2Department of Chemical and Biochemical Engineering, Tecnológico Nacional de México—Instituto Tecnológico de Durango (TecNM-ITD), Durango 34080, Mexico; 3Tecnológico de Monterrey, School of Engineering and Sciences, Monterrey 64849, Mexico

**Keywords:** microalgae, macroalgae, cosmetology, photoprotectans, UV radiation, biomass

## Abstract

In the last decade, algae applications have generated considerable interest among research organizations and industrial sectors. Bioactive compounds, such as carotenoids, and Mycosporine-like amino acids (MAAs) derived from microalgae may play a vital role in the bio and non-bio sectors. Currently, commercial sunscreens contain chemicals such as oxybenzone and octinoxate, which have harmful effects on the environment and human health; while microalgae-based sunscreens emerge as an eco-friendly alternative to provide photo protector agents against solar radiation. Algae-based exploration ranges from staple foods to pharmaceuticals, cosmetics, and biomedical applications. This review aims to identify the effects of UV and UV-vis irradiation on the production of microalgae bioactive compounds through the assistance of different techniques and extraction methods for biomass characterization. The efficiency and results focus on the production of a blocking agent that does not damage the aquifer, being beneficial for health and possible biomedical applications.

## 1. Introduction

Healthy coral reefs are one of the most valuable ecosystems on earth as they provide food and coastal protection; however, some chemicals or active ingredients within sunscreens can cause permanent damage to coral reefs [1]. The most known harmful chemical compounds are oxybenzone and octinoxate; nonetheless, other chemicals included in commercial sunscreens are benzophenone-1, benzophenone-8, OD-PABA, 4-methylbenzylidene camphor, 3-benzylidene camphor, nano-titanium dioxide, and nano-zinc oxide [2,3,4]. Cosmetic ingredients are highly regulated to assure safety and efficiency standards, besides the fact that not allowed compounds may be present in a cosmetic at trace levels due to processing conditions, the prevalence of synthetic ingredients over naturals is related to the fact that the final composition of a commercial cosmetic must satisfy the corresponding legislation [5]. It is common practice that a synthetic compound may be chosen over a natural source due to the stability of the molecule of interest. However, a natural form of astaxanthin (a microalgal pigment of current interest) may compete against the synthetic molecule, not only due to its safety and reduced environmental stress but also due to the higher antioxidant activity related to the esterified form in comparison to synthetic astaxanthin. Related to this condition high-value bioactive molecules from microalgae such as astaxanthin are being explored for pharmaceutical and cosmetic products [6,7,8]. The formulation of sunscreens based on microalgae may have the necessary protection but need proper carriers due to the solubility of MAAs; they are not suitable for being on the skin for a long period. It has been reported that the use of microalgae extracts is beneficial for the ecosystem and human beings. Such extracts are useful in the formulation of skin care products offering daily sun protection that oils in other formulations do not have. However, there is a need to improve the formulation to carry, deliver, and extend the retention period on the skin [9,10]. This work looks to report bioactive compounds from algae with UV-protection activity and new methods to formulate longer-lasting topic formulation.

## 2. Solar Radiation

The level of solar radiation that reaches the earth’s surface has increased dramatically in recent years due to the decrease and changes in the permeability of the ozone layer. The UV index published by the Environmental Protection Agency of the United States (EPA), presents projections of the risk of overexposure to UV radiation; just as an example of the severity of the change in radiation levels, in 1994 in New York City, the forecast for the number of days per year with high UV exposure risk was 29, with no expected days with a UV index of very high risk, but in 2017, the number of days with high UV exposure prediction was 66, and even 58 days were predicted to feature very high levels of UV radiation [11]. With this data in mind, the current importance of photoprotection becomes obvious. Ultraviolet radiation is defined as the electromagnetic energy emitted at wavelengths shorter than those corresponding to that visible to the human eye, but greater than what characterizes X-rays, between 100 and 400 nm. UV radiation is biologically harmful, damaging the DNA of cells, and it can cause genetic defects on external surfaces if received in high doses. UV radiation can damage human skin, causing slight redness to burns over time. It can cause serious discomfort, moles, blemishes, and even skin cancer; when the body perceives damage from sunlight, it sends it to the affected cells to avoid further damage and darkens the skin. It has harmful effects in the short and medium term. The reddening of the skin (solar erythema), from mild to severe burns, is the main immediate harmful effect. The medium-term effects include the most frequent skin cancers and premature aging of the skin and changes in the DNA of living beings [12,13]. Depending on the wavelength, UV radiation can be classified into three categories, UVA, UVB, and UVC.

Occupational exposure to solar radiation is a current problem, as well as the geographic variation of UV exposure; these environmental characteristics determine risk levels that concern occupational safety and health departments [14]. There are six phototypes according to their UVR skin response; melano-compromised (1), melano- compromised (2), melano-competent (3), melano competent (4), melano-protected (5), and melano protected (6). In this case, the skin responses to UVR radiation, correspondingly, are: always sunburns with no tan, often sunburns with light tan, sometimes sunburns with medium tan, rarely sunburns with dark tan, rarely sunburns with naturally dark skin, and no sunburns with naturally dark skin. These phototypes are according to a Standard Erythemal Dose (SED) of 100 J/m^2^ [14].

UVA radiation is responsible for tanning, and it is continuously visible radiation, varying between 400 and 320 nm. UVB radiation is very dangerous for life when exposed for long periods, being the cause of many skin and eye diseases (skin, cancer, carats, etc.), it varies between 320 and 280 nm. UVC radiation is the most dangerous for human life. It is absorbed by the ozone layer and is below 280 nm, causing irritation, high-grade burns, and continuous damage to the skin. Figure 1 shows the harmful effects caused by solar radiation on human skin [15].

## 3. Algae and UVA Stimulation

UVA or long wavelength radiation (315–400 nm) is not absorbed by the ozone layer and constitutes 90% of the total radiation that reaches the earth’s surface. UVA rays are also known as “aging rays”, capable of passing through the dermis and are not only inducing the appearance of wrinkles but are also associated with the development of skin pathologies and the formation of reactive oxygen species (ROS) [16]. Studies by Huang et al. in 2018 about the UVA radiation induction of microalgae metabolites reported the increase in total xanthophylls and total mycosporine-like amino acids through a 365 nm UVA light with a culture of *Nitzschia closterium* and *Isochrysis zhangjiangensis* [17]. Additionally, Huang et al. in 2018 investigated the effects of microalgae against UV radiation and explored fecosterol mechanisms in microalgae such as *Nitzschia closterium* by using classical methods including a semi-quantitative reverse transcription polymerase chain reaction, HPLC (high-performance liquid chromatography, reducing the concentration of UV-induced matrix metalloproteinases (MMPs), and inflammation caused by cytokine expression. Fucoxanthin is a carotenoid that is a potent antioxidant because it reduces (ROS) and MMP-13 expression in addition to inhibiting UV-induced VEGF expression in the skin. Astaxanthin is a carotenoid derived from algae; it is widely used as an ingredient in skin care products for its immune system-stimulating properties. Álvarez-Gómez et al. in 2016 found that fucoidan inhibits melanin synthesis by down-regulating melanogenesis-associated transcription factor (MITF) and protein tyrosinase. This result suggests that fucoidan can be used as an anti-pigmentation ingredient in medical and cosmetic fields. It has been described with antioxidant and anti-angiogenic properties. Below, Table 1 shows different compounds that are induced by exposure to UVA irradiation of macro- and microalgae understanding macroalgae as a multicellular form of an organism and microalgae as a unicellular form [18,19].

## 4. Algae and UVB Stimulation

Surface UVB irradiation (280–315 nm) is related not only to the stratospheric ozone layer but also to other tropospheric factors (aerosols, clouds, etc.), surface albedo, sea ice, and snow [21]. As shortwave solar radiation, UVB irradiation can penetrate 20 to 30 m into the ocean water and cause serious deleterious effects on marine organisms. Therefore, there has been great concern about the increase in UVB irradiation and its consequences in the biosphere throughout the 21st century, although the ozone layer has improved in recent years due to the implementation of the Montreal protocol [22]. Phytoplankton, similarly, to chlorophyll, is subjected to stress induced by UVB irradiation since they live in a habitat where UV radiation has been changing. In general, high doses of UVB irradiation have been widely reported to cause stress to microalgae, resulting in DNA damage, photosynthesis impairment, lipid peroxidation, cyclobutene pyrimidine dimer formation, and growth inhibition [23,24]. Although several physiological and biochemical studies have investigated the effects of UVB irradiation on phytoplankton, few studies have explored the impacts of UVB irradiation on algae community composition. Substantially reduce the rate of CO_2_ assimilation and the amount of rubisco and decrease the absorption of nutrients such as phosphorus and nitrate [25]. Below, in Table 2, the effects of UVB irradiation on microalgae are listed, showing the different types of microalgae and the compounds they produce. 

## 5. Algae and UVC Stimulation

The UVC, or short wavelength, radiation (100–280 nm) is of higher energy but is completely absorbed by oxygen and ozone in the atmosphere [26,27]. Continuous exposure to UV irradiation contributes to aging and aesthetic problems, with consequences such as sunburn, hyperpigmentation, or skin cancer. Several studies have shown that UVC irradiation causes a negative response in cellular processes, metabolism, and growth [28,29]. Despite the above, based on the knowledge that it is possible to obtain a beneficial effect with the sublethal application of an agent capable of inducing physical stress or chemical, it has been verified that controlled radiation with UVC in some organisms, using doses between 2 and 14 Kj/m^2^ can produce positive responses like in seedlings, fruits, and other plant structures in the postharvest [30,31]. Previous studies have found that UVC irradiation could inhibit the growth as well as the release of microcystins from *Microcystis aeruginosa* in artificial culture. In addition, UVC irradiation could cause marked damage to various targets of algae cells including the gene, photosynthesis system, cell membrane integrity, the inhibition of toxin production, and release. *M. aeruginosa and C. vulgaris* were chosen as cyanobacteria and microalgae to investigate the suppression effects of microalgae against UVC irradiation, looking for its photoprotective effect against solar radiation [32]. It has been reported that the ultrafiltration method is effective at a dose of 37–150 Kj/m^2^ and 0.180 Kj/m^2^ in the development of photoprotective metabolites ROM (reactive oxygen metabolites), resulting in antioxidant and protective effects. In Table 3 are gathered known compounds produced from microalgae as defense mechanisms against UVC irradiation.

**Table 2 molecules-27-05334-t002:** Algae compounds obtained by UVB irradiation.

Algae	Type of Algae	Compounds	Extraction Method	Yield (%)	Dose (Kj/m^2^∗d)	Cell Protection Effect	Reference
*Alexandrium catenella*	Microalgae	Diatoxanthin	HPLC	NR	NR	Change in concentration of photosynthetic pigment.	[33]
*Characium terrestre* *Coelastrum microporum* *Enallax coelastroides* *Enallax* sp. *Scenedesmus* sp. *Scotiella chlorelloide*	Microalgae	Porphyra-334	HPLC	NR	NR	Non-lethal UVB dose increase MAAs synthesis.	[34,35]
*Nannochlorop-sis oceanica* *Dunaliella salina* *Nannochloropsis limnetica*	Microalgae	Carotenoids and vitamin D_3_	CG	24 ± 0.1	3–6	Accumulation of α-tocopherol and β-carotene	[36]
*Rhodomonas salina*	Microalgae	Carotenoids	HPLC	24.1 ± 0.1	16–22	Photoreactivation mechanism in cells to survive UVB damage.	[37]

HPLC: High-performance liquid chromatography; GC: Gas chromatography.

## 6. Influence of Light on Microalgae Growth and Bioproducts

Visible-ultraviolet radiation (UV-vis) is a photon emission of visible spectrum regions, between ultraviolet and infrared (400–780 nm).

UV-vis radiation uses electromagnetic radiation in the visible regions. It is used to identify some functional groups, in this case seaweed, such as temperature, salinity, or CO_2_ concentration. Microalgae can survive different doses of UV-vis radiation in extreme conditions since their physiology is capable of changing, promoting their adaptation through the production of secondary metabolites that allow them to conquer different environments. The intensity of UV-vis radiation can exert a harmful effect on the photosynthetic process and on the cellular components of algae, but it also promotes protection and repair mechanisms against UV radiation. One of the main protection mechanisms is the biosynthesis and accumulation of molecules such as carotenoids and phycobiliproteins.

Microalgae and macroalgae are phototrophic organisms, where their primary and secondary metabolism strongly depends on lighting for their growth and the production of biomass and secondary metabolites, which has generated significant efforts to achieve adequate irradiation (lighting time, intensity, and wavelength) to achieve more efficient production of the metabolite of interest [40]. The term chromatic acclimation proposed demonstrated all the mechanisms of algae to optimize light capture in response to variations in light irradiation variations [41]. The effect of light on the growth and production of microalgal metabolites has been shown to depend on the species and genetics of each microorganism, with no possibility of transfer from one strain to another [42].

*Arthrospira maxima* and echinenone were examined under UV-vis radiation at a dose of 400–490 nm to look for the different metabolites and effects caused by this process. Previous studies found that *A. maxima* develops secondary metabolites capable of generating photoprotective agents that are useful for the protection of microalgae from UV-vis radiation and block photochemical reactions in the epidermis. Spirulina is the commercial name for the edible cyanobacteria of the genus *Arthrospira* that contains high protein content, fatty acids, vitamins, and minerals [43,44]. It was examined by the high-power elevation system at a dose of 300–500 nm to identify the production of photoprotective agents, and it was concluded that improvement in the interaction of compounds with the polymer due to the presence of hydroxyl in its structure provided layers of protection against light [45].

Different studies have shown that the joint supply of monochromatic lights provides more energy for the photosynthesis of microalgae cultures. The combination of blue and red light increases the speed with which microalgae reach the stationary phase, through lower doubling rates and generation times. The simultaneous application of red and blue light in *Dunaliella salina* showed biomass growth and an increase of up to 35.33% in lipid production, and blue light also showed significant increases in protein, carbohydrate, and lipid production [46].

Similarly, Severes et al. (2017), demonstrated in cultures of *Chlorella* sp. that the mixture of red 660 nm and blue 465 nm wavelengths in a 1:1 ratio induces an increase in chlorophyll content up to 3.64 μg/mL, compared to 2.62 μg/mL obtained with red light. The combination of red:blue light in proportions of 5:5, 7:3, and 3:7 induced an increase in biomass production up to 494, 456, and 452 mg/L, respectively, against 300–330 mg/L obtained with monochrome light [47].

The absence of blue light and the presence of green light significantly downregulated genes involved in carbon fixation and decreased photosynthetic efficiency in *Nannochloropsis gaditana*, converting β-carotene as the main derivative pigment to low nutrient levels, mimicking the conditions of the absence of nutrients [37,40].

The maximum productivity of lipids in different cultures of microalgae strains has been achieved with light intensities ranging from 60 to 700 µmol photons m^−2^ s^−1^, the use of red and blue light in different proportions, as well as the photoperiod or pulsed light, achieving a regulatory effect on the synthesis of fatty acids and carotenoids such as lutein, β-carotene, and astaxanthin [48].

The strategic management of different monochromatic wavelengths allows to achieve more sustainable cultures of photoautotrophic strains, through the first stage of growth and the accumulation of biomass at favorable wavelengths, followed by a stage of an abrupt change to an unfavorable wavelength, to induce the production of secondary metabolites such as lipids and bioactive compounds. Table 4 list some compounds produced from microalgae as defense mechanisms against UV-vis irradiation.

## 7. UV Protective Metabolites

### 7.1. Mycosporine-like Amino Acids

MAA are found naturally in seaweed, and a high MAA content has been shown in red algae and brown algae. The structure of MAAs includes cyclohexanediones or cyclohexene aldehydes in the active site that are linked to another active group through the phenol hydroxyl. MAAs are hypothesized to absorb light at a wavelength of 320 to 360 nm; therefore, MAAs exhibit a strong ultraviolet absorption capacity. In this research, MAA from *Porphyra haitanensis* was extracted, separated, and purified and the mechanisms responsible for its cutaneous anti-photoaging effect and as explored by using a mouse cutaneous photoaging model. These experiments, involving local treatments and washes administered with various doses of MAA, were effective against skin photoaging, and metalloproteinases (MMPs) content and expression were detected in skin tissue homogenates, combined with the pathological analysis of the anti-photoaging activity and underlying mechanisms of MAAs. The anti-photoaging activity and underlying mechanisms of MAAs were analyzed. Scytonemin is another UVA-, UVB-, and UVC-filtering compound; this is a highly stable, yellowish-brown, lipid-soluble, inducible pigment. Scytonemin is found exclusively in the polysaccharide sheath of some cyanobacteria as a protective mechanism. During periods of desiccation, scytonemin becomes more important as a UV protector due to the inactivation of other detection mechanisms. Its UV protection properties mean scytonemin is a powerful sunscreen material [48,54]. Exposure of *Chlamydomonas nivalis* to ultraviolet light induced the production of bioactive compounds with antioxidant properties. The tolerance to UV rays of the snow algae *C. nivalis* and the ability to produce under this radiation phenolic compounds, free proline, and antioxidant protection factors in response to UV-A and UV-C light generates a great potential for biotechnological and pharmaceutical application [55,56].

### 7.2. Carotenoids

Carotenoids are known as fat-soluble plant pigments widely distributed in nature that provide various colors such as yellow, red, and orange to fruits and vegetables. They are synthesized mainly by plants and algae, as well as by fungi and bacteria. Carotenoids can be found throughout the animal kingdom and in humans due to selective absorption throughout the food chain [57]. These lipophilic molecules are based on the chemical structure classified as carotenes and xanthophylls, and both classes have a common C40 polyisoprenoid structure containing a series of centrally located conjugated double bonds that act as a light-absorbing chromophore. Carotenoids that exist as pure nonpolar hydrocarbons are called carotenoids (α-carotene, β-carotene, and lycopene); on the contrary, xanthophylls (β-cryptoxanthin, lutein, zeaxanthin, and astaxanthin) are more polar carotenoids that contain oxygen as a functional group in their structure, either as a hydroxyl group or a keto group as a terminal group [58]. So far, more than 800 carotenoids have been identified, but only several are found in the human body, including α-carotene, β-carotene, lutein, and lycopene, as well as zeaxanthin and α- and β-cryptoxanthin. People are constantly exposed to ultraviolet (UV) light, some less, some more, depending on where they live, their activities, what they do for a living, hobbies, culture, but also their understanding of the importance of sun protection and its implementation. Exposure to solar ultraviolet light has been estimated to be 10% of the total available annual UVR for outdoor workers and 3% for adults working indoors [58,59]. It is crucial to keep in mind that sunlight stimulates blood circulation and bone health, since the exposure to some sunlight induces vitamin D production in the human body. According to the World Health Organization (WHO), 5 to 15 minutes of sun exposure per week is enough to maintain healthy vitamin D levels in the body [60]. In fact, as the UV Index tends to be higher with closeness to the equator, sunlight exposure must be preferably less for countries nearby this area [61].

## 8. Biomedical Applications

Human skin is the largest organ in the integumentary system that covers the entire body surface. The skin is a complex organ consisting of three primary layers: the epidermis, the dermis, and the hypodermis. The epidermis is the outermost layer of the skin, which plays a protective role against environmental damage and is resistant to water. The epidermis has no blood vessels, and the main cells contain keratinocytes (content around 95%), melanocytes, Merkel cells, and Langerhans cells. The epidermis could be subdivided into three cell layers, the upper one being a superficial corneal layer, which is composed of flattened cells that contain the proteinaceous and resistant factors keratin [62]. Since the skin interacts directly with the environment, it is sensitive to stimuli and can even receive damage from chemical and physical substances, especially ultraviolet (UV) radiation [63]. Continuous radiation exposure often has many complications, including sunburn, hypopigmentation, and even skin cancer. However adequate levels of sunlight are necessary for correct human body function, besides the production of D vitamin and their repercussion on rickets and osteoporosis prevention. Diseases such as lupus vulgaris (skin tuberculosis) were successfully cured with UVB stimuli. Psoriasis, an autoimmune disease, can be treated with UVA radiation, and the same with vitiligo [60].

### 8.1. Algae against Acne

Bioactive compounds extracted from seaweed could be a natural and safe alternative. Macroalgae extracts have been reported to possess antibacterial properties and antifungal activities. Ruxton and Jenkins (2015) report a new algal zinc-oligosaccharide complex (SOZC) from the polysaccharide membrane of *Laminaria digitata* through a series of double-blind, placebo-clinical trials. The findings suggest that SOZC may relieve acne symptoms [64].

### 8.2. Algae Protect the Skin from Damage by UV Radiation

Photoaging caused by excessive exposure to sunlight has become a huge problem in recent years. Marine organisms, especially. The bioactive compounds of macroalgae can absorb UVA and UVB; some of them eliminate ROS and inhibit the formation of MMP. Various extracts from different algae exhibit photoprotective functions [65,66]. Compounds in those extracts that have confirmed photoprotective activity include shinorine, Porphyra-334, palythene, eckstolonol, eckol, Mycosporine-glycine, Mycosporine methylamine-serine, sargacromenol, fucoxanthin, tetraprenyltoluquinol chromane meroterpenoid, tetraprenyltoluquinol chromane meroterpenoid, and sargaquinoic acids [67]. The most efficient UV absorber compounds are MAA, which are water-soluble substances found in many organisms, such as cyanobacteria and algae. Porphyra-334 can downregulate intracellular UV-activated ROS and controls MMP expression by eliminating damaged HDF overdoses [68]. Therefore, the search for safe and effective skin whitening agents from seaweed can be beneficial for the cosmetic industry. To find new anti-browning and bleaching agents, scientists screened various seaweeds for tyrosinase inhibitors and found some potential algae. Several species of microalgae are exploited in the cosmetic industry, especially in the skincare market, the main ones being *Arthrospira* and *Chlorella* [69]. The phlorotannins, active ingredients of nutraceuticals, are the most abundant polyphenols found in brown marine algae; their health applications are due to antioxidant activity [70,71]. Polysaccharides have been shown to have several important properties. However, attempts to establish a relationship between polysaccharides’ structures and their bioactivities have been challenging due to the complexity of this type of polymer. The polysaccharides produced by algae are presented according to their group of macroalgae, phaeophytes, rhodophytes, and chlorophytes, which are relative to microalgae. However, there are always some similarities between the polysaccharides of each group of algae: fucoidans are often extracted from brown algal species, agaroides come from red macroalgae, and ulvans are obtained from green algae [72].

## 9. Discussion

The problem of the entire cosmetology industry is the development of sunscreens without full knowledge about their use, compatibility, performance, and photostability. It has been detected that different chemical substances such as octyl methoxycinnamate and benzophenone-3 are used solely to increase the sun protection factor (SPF) levels in sunscreens. This is not possible, and they want to do it only by using these two sunscreens but using a higher concentration in their formula. This is serious, since in the case of octyl methoxycinnamate, the maximum concentration of use allowed worldwide is 10%, and with this concentration, only an SPF of 8 of protection can be achieved. In the spectrum of UVB radiation and with benzophenone-3, the maximum concentration allowed worldwide is 10%, and more than 90% of customers who are currently developing sunscreens still have the belief that this agent protects from UVA radiation. Additionally, their harmful effects in marine environments have been reported. Those effects emerge by accumulation in swimming pools and reactions with chlorine-producing hazardous by-products along with municipal, residential, and boat/ship wastewater discharge producing coral bleaching due to the endocrinal disruptor direct effects of oxybenzone [73,74,75]. Since the absorption spectrum of benzophenone-3 barely touches the spectrum of UVA radiation (320 to 400 nm), this harms the consumer.

On the other hand, the use of marine algae as sunscreens is more efficient, durable, and biodegradable than other chemical components traditionally used, which contribute less contamination of the water and also contribute to greater efficiency in the absorption of ultraviolet rays, which provides better protection against different ailments. The possibility of including MAAs molecules that naturally contain algae in sunscreen creams has been shown to allow them to be more effective and last. It has been confirmed that the stability of MAAs lasts up to 270 °C, which represents the stability of 100% after 6 h under the sun and can increase the degree of photoprotection. Otherwise, legislation is needed to include algae-derived photoprotectans metabolites as part of the GRAS list. This includes several assays to guarantee their safety as an active ingredient in sunscreen formulation.

The information available on the protective effects of microalgae on human skin suggests that they can be implemented in the dermatological field as cosmetics and sunscreens since some of the main advantages are that they stimulate and improve blood circulation, revitalize and firm the skin, they are toning, they are rebalancing, detoxifying, and naturally moisturizing. Red algae are used for acne treatments, meanwhile, brown algae for masks, creams, shampoos, or lotions. Greenish-brown algae are good for obesity treatments and green algae relieve stress and can be used for toners, creams, and exfoliants. Green algae are enriched with calcium, iron, potassium, magnesium, phosphorus, and vitamins A, B, C, E, F, and K. Microalgae are useful to protect the population from the damage caused by UVA, UVB, UVC, and UV-vis radiation with high risks such as outdoor works, high-grade burns, skin cancer, etc. The Department of Medicine and Dermatology at the University of Malaga and the Department of Photochemistry at the University of La Roja have developed compounds that provide greater stability and duration to sunscreens. They have been inspired by substances produced by certain fungi and algae naturally. It offers a new formulation with greater guarantees against the sun. Researchers have confirmed the absence of allergies and a longer duration of its effect to avoid application several times a day during sun exposure.

Processes for purification and extraction became a challenge due to the nature of the different bioactive metabolites and the strain characteristics. New, sustainable protocols are needed to improve raw biomass processing [76]. As a result of microalgae bioremediation processes such as CO_2_ phycocapture, water treatment, and phycopigments production, microalgae biomass will be generated, and its content properties can be used to develop application products that allowed the microalgae production and processing to be profitable to implement [77,78,79,80].

## 10. Conclusions

This work explores the protective mechanisms of microalgae against different types of radiation (UVA, UVB, UVC, and UV-vis). Radiation is a growing problem for human beings; its effects are of a greater degree which causes damage to the skin and health of living beings. The effects on marine algae and their characteristics, including their secondary metabolites, can be used as photoprotective agents against UV radiation. Microalgae have been shown to inhibit photoaging properties, and they can potentially work to prevent skin damage. Developing compounds based on microalgae is a good option for its multiple benefits on the skin (tonifying, detoxifying, etc.), helping to prevent diseases caused by long periods of light exposition such as skin cancer, and burns. The importance of this type of compound based on micro and macroalgae is key for medical use and the production of bio-based materials in life sciences such as pharmaceuticals, cosmetics, and medical devices. This research looks to resume the effects of UV and UV-vis radiation on the production of different compounds at different doses in algae species. Light is an essential requirement for photoautotrophic growth and establishing the spectral composition, light intensity, time, and frequency of illumination. It is a powerful tool to regulate the growth of microalgae biomass and induce the production of value-added metabolites for the development of photobioprotectors and biomedical products.

## Figures and Tables

**Figure 1 molecules-27-05334-f001:**
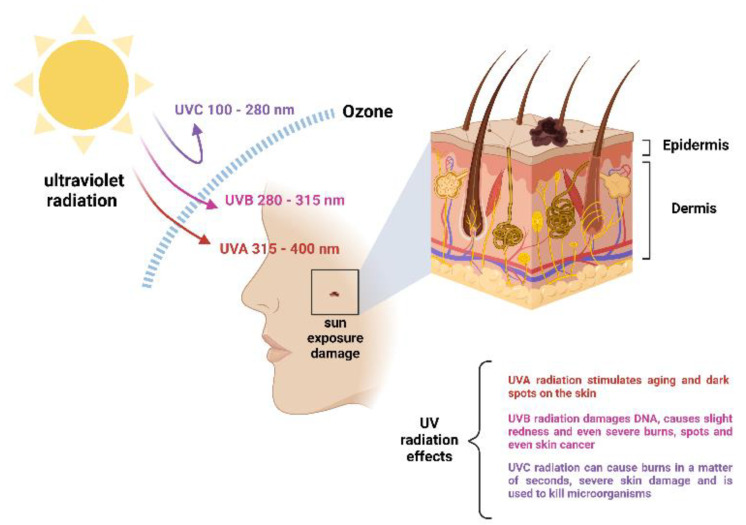
Scheme of types of UV radiation and its penetration into the skin. Figure was created with BioRender.com.

**Table 1 molecules-27-05334-t001:** Algae compounds obtained by UVA irradiation.

Algae	Type of Algae	Compounds	Extraction Method	Yield	Dose (W/t∗d)	Cell Protection Effect	Reference
*Isochrysis zhangjiangenesis*	Microalgae	Fucoxanthin, xanthophyll	HPLC	123 ± 5 mg g^−1^	10–13	Enhancement of bioactive microalgae metabolites including carotenoids and MAA efficiently.	[7]
*Nitzschia closterium*	Microalgae	Porphyra-334, chinorina	BBM discoloration	95–99%	21	It was shown as an activator of the cytoprotective pathway, demonstrating the potential against UV damage to DNA.	[14]
*Chlorella* sp.	Microalgae	MAA	Solid phase extraction	50%	80	Synthesis of MAA with a more efficient absorption spectrum.	[17]
*Porphyra umbilicalis*	Macroalgae	Palythine, palythinol, shinorine, porphyra-334, and asterina-330	With various alcoholic and hydroalcoholic solvents	2.0–2.6 mg	NR	Repair the damage to proteins caused by UV irradiation.	[18]
*Grateloupia*	Macroalgae	Porphyra-334	Radical ABTS	80–82%	70	Synthesis of MAAs with more efficient absorption.	[20]

HPLC: High-performance liquid chromatography; BBM: β-carotene bleaching method; ABTS: Oxidant agent of different compounds.

**Table 3 molecules-27-05334-t003:** Algae compounds obtained by UVC irradiation.

Algae	Type of Algae	Compounds	Extraction Method	Yield (%)	Dose	Cell Protection Effect	Reference
*Chlamydomonas reinhardtii*	Microalgae	4-hydroxybenzoic acid, catechin, and Chalcone Isomerase	Liquid–solid extraction	1.21	0.1 (W/m^2^)	Non-enzymatic ROS scavenging	[38]
*Nannochloropsis* sp.	Microalgae	Omega-3	PHEW	30	100 or 250 (mJ/cm^2^)	Antioxidant	[39]

PHEW: Ultrafiltration method, DW: Dry weight biomass.

**Table 4 molecules-27-05334-t004:** Algae compounds obtained by UV-vis radiation stimulation.

Algae	Type of Algae	Compounds	Extraction Method	Yield (%)	Dose (nm)	Mechanism of Action	Reference
*Spirulina platensis*	Microalgae	Carotenoids hydrocarbons	HPLC	96	400–490	It inhibits ERON and singlet oxygen as a photoprotective agent and blocks photochemical reactions in the epidermis.	[49]
*Haematococcus pluvialis*	Microalgae	Dihydroxy carotenoid diacyl esters	HPLC	97	400–490	Protection against UV radiation damage on microalgae.	[50]
*Phormidium autumnale*	Microalgae	cis-echinenone Carotenoids	HPLC	98	490	Increases the activity of polymorphonuclear cells (PMNs), promotes the release of lymphokines by lymphocytes, and increases the cytotoxic power of macrophages.	[51,52]
*Spirulina pacifica*	Microalgae	β-Cryptoxanthin Monohydroxy carotenoids	HPLC	95	490	Protection against damage caused by high amounts of UV radiation.	[53]

HPLC: High-performance liquid chromatography.

## Data Availability

Not applicable.

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
