# Peer review of "Effects of UV and UV-vis Irradiation on the Production of Microalgae and Macroalgae: New Alternatives to Produce Photobioprotectors and Biomedical Compounds"

_molecules, 2022, doi:10.3390/molecules27165334_

Round 1

Reviewer 1 Report

Generally, this article was well prepared, and useful for further utilization of algae to produce the medical products, such as sunscreens, by exposing to optimal UV and UV-vis radiation. However, there were some problems need to be revised. The detailed comments were as follows.

1.       The abbreviation of sth should be given when it first appearances. Such as reactive oxygen species (ROS) in line 89.

2.       Line 95: The description of “[8] investigated…” was not common in paper. I think [8] shouldn’t appear in the first place in a sentence.

3.       Line 124: CO2 should be CO2.

4.       Line 107, 128, and 151, The UV should be UVA, UVB, and UVC, respectively. Please revise. As shown in the headings of Table 1-3.

5.       Line 232, organ or organism?

6.       Line 242-248, These statements appear to repeat the previous text.

7.       Line 288, which table? Table 7?

8.       Line 291-293 seem hard to understand.

9.       I think the section 6 should be more detailed, dividing the UV-vis into red, orange, yellow…. And describe the corresponding effects on metabolites.

10.    The title should be revised. In the context, UV-vis was designed in line 160-161. But the authors mainly described the effects of UVa-c effects on metabolites in microalgae and macroalgae.

11. The discussion should be more detailed and references were required.

Author Response

Specific comments:

1. Lines 35-44 The problem which should be raised is that cosmetics producers prefer the chemical ingredients because  they are more stable in comparison to natural ones and more effective. Could authors comment the fact in the article.

Response: Thanks. Corresponding information was added in the introduction.

Line 44-46 Are authors sure? What about: UV protecting pigment of the terrestrial cyanobacterium Tolypothrix byssoidea, , A New UV-A/B Protecting Pigment in the Terrestrial Cyanobacterium Nostoc commune1, UV protection in desert cyanobacteria; Antarctic cyanobacteria?

Response: Thanks you are correct. Corresponding information was added in the introduction and the unclear information was deleted.

Line 59 Please put some concrete measurements, data from literature here.

Response: Thanks you are correct. Corresponding information was added in the introduction and the unclear information was deleted.

Line 61: There are two problems in dependence of geographical region people less exposed to sun became much more sensitive like in Northern Europe, scandinavian countries  and people overexposed like in tropical regions please make discrimination and differentiate reactions of organism. 

Response: Thanks. Pertinent information was added but keeping in mind occupational exposure and phototypes.

Line 92: this sentence not. clear   and change by Wang et al. .......

Response: Thanks. The sentence was corrected.

6. Line 95: do not begin sentence as [8] , give the name of author

Response: Thanks. The sentence was corrected.

Line 98 explain the shortcut ,,MMP’’

Response: Thanks. information added (matrix metalloproteinases)

Line 103 Please pay attention how to begin the sentence this is not acceptable

Response: Thanks. The sentence was corrected.

Line 107 Authors mean ,,In the table are collected information about organisms and compounds which biosynthesis is induced by UV?’’ please clarify this.
Response: Thanks. Comment attended

142 146 please make this fragment more clear.

Response: Thanks for the suggestion, the text was corrected.

11. line 147 this is not clear at all what authors want to communicate?

Response: Thanks. Corrected

Line 150 In table 3 are gathered known compounds……

Response: Thanks for the suggestion, the text was edited to fit the comment.

13. Comment Tables: please locate the tables in corresponding paragraphs. 

Response: Thanks. Tables location were attended.

Table 1 column: Mechanism of action: In this column there is no precise information as in the column tittle? please verify it.

Response: Thanks. The column title was adequate.

15. Type of algae:  Algae -Eukaryotic; Cyanobacteria - Prokaryotes, please discriminate properly. Define micro and macroalgae and use it properly in your document.
Algae species please give full names in all rows and look on the proper grammar and proper name me quedé en T7 ex. Isochrysis zhangjiangensis, Alexandrium catenella check all of them
Response: Thanks. Names were corrected, and the concept of macro an microalgae defined and employed repectively in the tables.

Row 1 both molecules have the same activity in the row below you have different informations?

Response: Thanks for the comment, the incoherent information was deleted and row 2 was corrected.

17. Row 2  verify the language french or english?
clearly divide the source of data
Response: Thanks. The row was corrected.

18. Saxitoxin - this is a neurotoxin, in mechanism of action - please more precisely what want you communicate. 

Response: the section was corrected.

19. Row 1 Microcystis aeruginosa that are cyanobacteria if it is different species give full name

Response: Thanks. The column was adequate.

20. Row 2 Toxic cyanobacteria - this concern the row 1? sin referencia
Response: the section was corrected.
21. Row 5 that are cyanobacteria rather Arthrospira maxima
Prophyra it is a genus name of red algae so please verify what compound was extracted from Arthrospira after UV induction sin referencia

Response: the section was corrected.

Line 172 : Arthrospira maxima and A. maxima

Response: Thanks. Corrected

Line 175 Pay attention Sprilina this is a name of commercial product see Furmaniak et all 2017 10.3389/fmicb.2017.02541

Response: Thanks. Corrected (Spirulina)

Line 185 correct: been shown

Response: Thanks. Corrected

25 .Line 197 Escitonema - Authors mean Scytonemin? please verify the proper name

Response: Thanks. Corrected (Scytonemin)

Line 207- 209  Could authors write this sentence different: There are known several types of MAA which differ in effects? …….(Table5).

Response: Thanks we appreciate the observation, the text was ambiguous. Corrected 

Comment: I think there is more  examples than you presented in this table.

Response: Thanks we appreciate the observation, some new information was added. 

Line 224 225 I propose to make this observation in comparison to the life style in well developed countries people much less time spend outside of buildings and therefore are probably more sensitive on UV thats is visible for instance for Scandinavian population. Please mention in your text that people need UV for health also in some doses.

Response: Thanks we appreciate the observation, the text improved with the corresponding information.

29. Line 229 Could you refer to the table as in the table 5  example?

Response: Thanks we appreciate the observation, the text was ambiguous. Corrected

There is richness of carotenoids in the nature, only selected examples are presented in table 6 (criterion of selection?) The biosynthesis is induced by UV radiation that authors mean?

Response: the section was corrected, table 6 was deleted.

Line 223:  Surface [75]. What do authors want to say?
Response: Thanks. Corrected, there was a dot between the sentence (that covers the entire body surface)

Chapter Biomedical applications : Please mention in your text that: In the appropriate amounts, UV radiation has a very beneficial effect on the human body. It improves well-being and increases mental and physical efficiency. It influences the secretion of melatonin and initiates the synthesis of vitamin D3, which is necessary e.g. to maintain normal bone physiology. As a result, it prevents rickets in children, and in adults - osteoporosis. In addition, it has been noticed that it increases the body's tolerance to skin grafts, and also promotes the treatment of many dermatological diseases. This is why ultraviolet radiation is often used in medicine. We have to keep it in mind.

Response: Thanks we appreciate the observation (the first part of the comment was included as part of another observation above) and the text was improved with the corresponding information according to this point.

Line 255, 266: are all of them mentioned in the tables?

Response: the section was corrected.

Line 288 you mean table 7?

Response: Thanks. Corrected

Line 290 there is lack a paragraph about problems encountered with applications of natural UV filters. and chalenges for the future research.

Response: The last paragraph in discution section was include to attend your recommendation.

36. Line 294: It sould be worth full to express that synhetic are environmentally dangerous and cite some data too, ex. Toxicopathological Effects of the Sunscreen UV Filter, Oxybenzone (Benzophenone-3), on Coral Planulae and Cultured Primary Cells and Its Environmental Contamination in Hawaii and the U.S. Virgin Islands.

Response: The mention report and others were included into discution section to extend the harmfull effects of those chemicals.

Line 295  explain FPS shortcut generally  all shortcuts must be explained.

Response: Thanks. Corrected. SPF (Sun protection factor).

line 303  show absorption spectra for these compounds.

Response: Thanks the absortion spectra was added to ilustrate the context. 

Line 304-307 this must be supported by the evidence from the literature

Response: Comment attended.
40. Line 308-309 please literature patents etc.

Response: we consider not include patents to focus in algae properties.

41. Line 313-314 the result of research on this field must be cited also.

Response : Favas, R., Morone, J., Martins, R., Vasconcelos, V., & Lopes, G. (2021). Cyanobacteria and microalgae bioactive compounds in skin-ageing: Potential to restore extracellular matrix filling and overcome hyperpigmentation. Journal of Enzyme Inhibition and Medicinal Chemistry, 36(1), 1829-1838.

Pangestuti, R., Siahaan, E. A., & Kim, S. K. (2018). Photoprotective substances derived from marine algae. Marine drugs, 16(11), 399.

Line 329 There are FDA-approves sunscreen ingredients from algae? could you verify and include the text that would be important

Response: FDA validation is need, this was include in discussion section. 

Reviewer 2 Report

General comment
There are some lacking informations which cause the article is not very actual they are listed in specific comments.

The tables need to be reconstructed actualised and precisely verified because some informations are mistakenly given and there are errors in nomenclature and taxonomic designations, names of compounds what make the reader confused.
The parameters given in the tables must be explained what authors want to communicate by parameter of yield efficiency of extraction or efficiency of energy absorption, or efficiency of UV induction?
All shortcuts must be defined.

I would also to stress that the tittle is: Effects of UV-vis radiation on the production of microalgae and
macroalgae: so I would expect some data concerning the level of induction under UV conditions if are examples accessible.

Specific comments:

Lines 35-44 The problem which should be raised is that
 cosmetics producers prefer the chemical ingredients because  they are more stable in comparison to natural ones and more effective. Could authors comment the fact in the article.

Line 44-46 Are authors sure? What about: UV protecting pigment of the terrestrial cyanobacterium Tolypothrix byssoidea, , A New UV-A/B Protecting Pigment in the Terrestrial Cyanobacterium Nostoc commune1, UV protection in desert cyanobacteria; Antarctic cyanobacteria?

Line 59 Please put some concrete measurements, data from literature here.

Line 61: There are two problems in dependence of geographical region people less exposed to sun became much more sensitive like in Northern Europe, scandinavian countries  and people overexposed like in tropical regions please make discrimination and differentiate reactions of organism.

Line 92: this sentence not. clear   and change by Wang et al. .......

Line 95: do not begin sentence as [8] , give the name of author

Line 98 explain the shortcut ,,MMP’’

Line 103 Please pay attention how to begin the sentence this is not acceptable

line 107 Authors mean ,,In the table are collected information about organisms and compounds which biosynthesis is induced by UV?’’ please clarify this.

142 146 please make this fragment more clear.
line 147 this is not clear at all what authors want to communicate?

Line 150 In table 3 are gathered known compounds…...

Comment Tables: please locate the tables in corresponding paragraphs

Table 1 column: Machanism of action: In this column there is no precise information as in the column tittle? please verify it.

Type of algae:  Algae -Eukaryotic; Cyanobacteria - Prokaryotes, please discriminate properly. Define micro and macroalgae and use it properly in your document.
Algae species please give full names in all rows and look on the proper grammar and proper name ex. Isochrysis zhangjiangensis, Alexandrium catenella check all of them

Table 1 row 1 both molecules have the same activity in the row below you have different informations?

row2  verify the language french or english?
clearly divide the source of data

TABLE2

Saxitoxin - this is a neurotoxin, in mechanism of action - please more precisely what want you communicate.

Table 3

row 1 Microcystis aeruginosa that are cyanobacteria if it is different species give full name

row 2 Toxic cyanobacteria - this concern the row 1?

row 5 that are cyanobacteria rather Arthrospira maxima
Prophyra it is a genus name of red algae so please verify what compound was extracted from Arthrospira after UV induction

line 172 : Arthrospira maxima and A. maxima

line 175 Pay attention Sprilina this is a name of commercial product see Furmaniak et all 2017 10.3389/fmicb.2017.02541

Line 185 correct: been shown
line 197 Escitonema - Authors mean Scytonemin? please verify the proper name
Line 207- 209  Could authors write this sentence different:
There are known several types of MAA which differ in effects? …….(Table5).
comment: I think there is more  examples than you presented in this table.

line 224 225 I propose to make this observation in comparison to the life style in well developed countries people much less time spend outside of buildings and therefore are probably more sensitive on UV thats is visible for instance for Scandinavian population. Please mention in your text that people need UV for health also in some doses.

line 229 Could you refer to the table as in the table 5  example?
There is richness of carotenoids in the nature, only selected examples are presented in table 6 (criterion of selection?) The biosynthesis is induced by UV radiation that authors mean?

Line 223:  Surface [75]. what authors want to say?

Chapter Biomedical applications : Please mention in your text that: In the appropriate amounts, UV radiation has a very beneficial effect on the human body. It improves well-being and increases mental and physical efficiency. It influences the secretion of melatonin and initiates the synthesis of vitamin D3, which is necessary e.g. to maintain normal bone physiology. As a result, it prevents rickets in children, and in adults - osteoporosis. In addition, it has been noticed that it increases the body's tolerance to skin grafts, and also promotes the treatment of many dermatological diseases. This is why ultraviolet radiation is often used in medicine. We have to keep it in minds.

Line 255, 266: are all of them mentioned in the tables?
Line 288 you mean table 7?

Line 290 there is lack a paragraph about problems encountered with applications of natural UV filters. and chalenges for the future research.

Line 294: It sould be worth full to express that synhetic are environmentally dangerous and cite some data too, ex. Toxicopathological Effects of the Sunscreen UV Filter, Oxybenzone (Benzophenone-3), on Coral Planulae and Cultured Primary Cells and Its Environmental Contamination in Hawaii and the U.S. Virgin Islands.

Line 295  explain FPS shortcut generally  all shortcuts must be explained.

line 303  show absorption spectra for these compounds
line 304-307 this must be supported by the evidence from the literature
line 308-309 please literature patents etc.
line 313-314 the result of research on this field must be cited also.
Line 329 There are FDA-approves sunscreen ingredients from algae? could you verify and include to the text that would be important.

Author Response

Specific comments:

1. Lines 35-44 The problem which should be raised is that cosmetics producers prefer the chemical ingredients because  they are more stable in comparison to natural ones and more effective. Could authors comment the fact in the article.

Response: Thanks. Corresponding information was added in the introduction.

Line 44-46 Are authors sure? What about: UV protecting pigment of the terrestrial cyanobacterium Tolypothrix byssoidea, , A New UV-A/B Protecting Pigment in the Terrestrial Cyanobacterium Nostoc commune1, UV protection in desert cyanobacteria; Antarctic cyanobacteria?

Response: Thanks you are correct. Corresponding information was added in the introduction and the unclear information was deleted.

Line 59 Please put some concrete measurements, data from literature here.

Response: Thanks you are correct. Corresponding information was added in the introduction and the unclear information was deleted.

Line 61: There are two problems in dependence of geographical region people less exposed to sun became much more sensitive like in Northern Europe, scandinavian countries  and people overexposed like in tropical regions please make discrimination and differentiate reactions of organism. 

Response: Thanks. Pertinent information was added but keeping in mind occupational exposure and phototypes.

Line 92: this sentence not. clear   and change by Wang et al. .......

Response: Thanks. The sentence was corrected.

6. Line 95: do not begin sentence as [8] , give the name of author

Response: Thanks. The sentence was corrected.

Line 98 explain the shortcut ,,MMP’’

Response: Thanks. information added (matrix metalloproteinases)

Line 103 Please pay attention how to begin the sentence this is not acceptable

Response: Thanks. The sentence was corrected.

Line 107 Authors mean ,,In the table are collected information about organisms and compounds which biosynthesis is induced by UV?’’ please clarify this.
Response: Thanks. Comment attended
142 146 please make this fragment more clear.

Response: Thanks for the suggestion, the text was corrected.

11. line 147 this is not clear at all what authors want to communicate?

Response: Thanks. Corrected

Line 150 In table 3 are gathered known compounds……

Response: Thanks for the suggestion, the text was edited to fit the comment.

13. Comment Tables: please locate the tables in corresponding paragraphs. 

Response: Thanks. Tables locations were attended.

Table 1 column: Mechanism of action: In this column there is no precise information as in the column tittle? please verify it.

Response: Thanks. The column title was adequate.

15. Type of algae:  Algae -Eukaryotic; Cyanobacteria - Prokaryotes, please discriminate properly. Define micro and macroalgae and use it properly in your document.
Algae species please give full names in all rows and look on the proper grammar and proper name me quedé en T7 ex. Isochrysis zhangjiangensis, Alexandrium catenella check all of them
Response: Thanks. Names were corrected, and the concept of macro and microalgae defined and employed respectively in the tables.

Row 1 both molecules have the same activity in the row below you have different informations?

Response: Thanks for the comment, the incoherent information was deleted and row 2 was corrected.

17. Row 2  verify the language french or english? clearly divide the source of data
Response: Thanks. The row was corrected.

18. Saxitoxin - this is a neurotoxin, in mechanism of action - please more precisely what want you communicate. 

Response: the section was corrected.

19. Row 1 Microcystis aeruginosa that are cyanobacteria if it is different species give full name

Response: Thanks. The column was adequate.

20. Row 2 Toxic cyanobacteria - this concern the row 1? sin referencia
Response: the section was corrected.
21. Row 5 that are cyanobacteria rather Arthrospira maxima
Prophyra it is a genus name of red algae so please verify what compound was extracted from Arthrospira after UV induction sin referencia

Response: the section was corrected.

Line 172 : Arthrospira maxima and A. maxima

Response: Thanks. Corrected

Line 175 Pay attention Sprilina this is a name of commercial product see Furmaniak et all 2017 10.3389/fmicb.2017.02541

Response: Thanks. Corrected (Spirulina)

Line 185 correct: been shown

Response: Thanks. Corrected

25 .Line 197 Escitonema - Authors mean Scytonemin? please verify the proper name

Response: Thanks. Corrected (Scytonemin)

Line 207- 209  Could authors write this sentence different: There are known several types of MAA which differ in effects? …….(Table5).

Response: Thanks we appreciate the observation, the text was ambiguous. Corrected 

Comment: I think there is more  examples than you presented in this table.

Response: Thanks we appreciate the observation, some new information was added. 

Line 224 225 I propose to make this observation in comparison to the life style in well developed countries people much less time spend outside of buildings and therefore are probably more sensitive on UV thats is visible for instance for Scandinavian population. Please mention in your text that people need UV for health also in some doses.

Response: Thanks we appreciate the observation, the text improved with the corresponding information.

29. Line 229 Could you refer to the table as in the table 5  example?

Response: Thanks we appreciate the observation, the text was ambiguous. Corrected

There is richness of carotenoids in the nature, only selected examples are presented in table 6 (criterion of selection?) The biosynthesis is induced by UV radiation that authors mean?

Response: the section was corrected, and table 6 was deleted.

Line 223:  Surface [75]. What do authors want to say?
Response: Thanks. Corrected, there was a dot between the sentence (that covers the entire body surface)

Chapter Biomedical applications : Please mention in your text that: In the appropriate amounts, UV radiation has a very beneficial effect on the human body. It improves well-being and increases mental and physical efficiency. It influences the secretion of melatonin and initiates the synthesis of vitamin D3, which is necessary e.g. to maintain normal bone physiology. As a result, it prevents rickets in children, and in adults - osteoporosis. In addition, it has been noticed that it increases the body's tolerance to skin grafts, and also promotes the treatment of many dermatological diseases. This is why ultraviolet radiation is often used in medicine. We have to keep it in mind.

Response: Thanks we appreciate the observation (the first part of the comment was included as part of another observation above) and the text was improved with the corresponding information according to this point.

Line 255, 266: are all of them mentioned in the tables?

Response: the section was corrected.

Line 288 you mean table 7?

Response: Thanks. Corrected

Line 290 there is lack a paragraph about problems encountered with applications of natural UV filters. and chalenges for the future research.

Response: The last paragraph in the discussion section was included to attend to your recommendation.

36. Line 294: It sould be worth full to express that synhetic are environmentally dangerous and cite some data too, ex. Toxicopathological Effects of the Sunscreen UV Filter, Oxybenzone (Benzophenone-3), on Coral Planulae and Cultured Primary Cells and Its Environmental Contamination in Hawaii and the U.S. Virgin Islands.

Response: The mentioned report and others were included in the discussion section to extend the harmful effects of those chemicals.

Line 295  explain FPS shortcut generally  all shortcuts must be explained.

Response: Thanks. Corrected. SPF (Sun protection factor).

line 303  show absorption spectra for these compounds.

Response: Thanks the absorption spectra were added to illustrate the context. 

Line 304-307 this must be supported by the evidence from the literature

Response: Comment attended.
40. Line 308-309 please literature patents etc.

Response: we consider not including patents to focus on algae properties.

41. Line 313-314 the result of research on this field must be cited also.

Response : Favas, R., Morone, J., Martins, R., Vasconcelos, V., & Lopes, G. (2021). Cyanobacteria and microalgae bioactive compounds in skin-ageing: Potential to restore extracellular matrix filling and overcome hyperpigmentation. Journal of Enzyme Inhibition and Medicinal Chemistry, 36(1), 1829-1838.

Pangestuti, R., Siahaan, E. A., & Kim, S. K. (2018). Photoprotective substances derived from marine algae. Marine drugs, 16(11), 399.

Line 329 There are FDA-approves sunscreen ingredients from algae? could you verify and include the text that would be important

Response: FDA validation is needed, this was included in the discussion section. 

Reviewer 3 Report

Radiation is a growing problem for humans, and current commercial sunscreens contain chemicals that are harmful to the environment and human health. In recent years, algae have attracted much attention due to their ability to produce bioactive compounds with photoprotective effects. This paper reviews the effects of different wavelengths of radiation on the production of bioactive substances in microalgae and macroalgae, providing new ideas for the biomedical application of algae and the production of photobioprotectors. However, there are some serious problems with this manuscript. I think the manuscript needs to be carefully considered before it is published. Here’re some specific questions and corresponding suggestions.

1.      What is the difference between UV, UV-vis and VIS? The descriptions of these terms in the title, abstract, keywords, and text of the article are inconsistent.

2.      Line 19, “bioactive compounds”, “photoprotective agents”, “carotenoids” and “mycosporine-like amino acids (MAAs)” are inclusion relationships rather than juxtaposition relationships.

3.      At the beginning of the introduction, the description is inconsistent with the references. Please check to see if the same problem exists in the rest of the manuscript.

4.      The Introduction section does not introduce enough background

5.      Lines 52-56, the sentence is too long, break it down into 2 sentences.

6.      Line 64, it seems that there is a pronoun problem here. Please change “which” to “those”.

7.      Figure 1, the wavelength range does not match the description in the text.

8.      The subject is missing in many statements in the text, such as line 95, line 103, line 147 and line 180, please check and correct them.

9.      Line 101,change “Astaxanthinis” to “Astaxanthin is”.

10.   At the end of chapters 3, 4 and 5, replace “UV radiation” with “UVA radiation”, “UVB radiation” and “UVC radiation” respectively.

11.   Line 121-128, the same sentence appears twice.

12.   Line 148, a space is required between the number “37-150” and the unit “Kj/m2”.

13.   Line 150, the pronoun “it” is missing after “support”.

14.   Table 1, are ABTS and BBM the extraction methods? And what’s the full name of BBM?

15.   Line 177, remove “a” before “high protein content”.

16.   In the section 7, the discussion on the species and activity of UV protective metabolites in algae is not complete.

17.   Line 185, a space is required between “been” and “shown”.

18.   Line 189-196, authors should avoid use of “we, our, us etc”.

19.   Line 202-207, check and improve the statement.

20.   What does Table 5 want to show? It seems to be inconsistent with the text.

21.   In section 7.2, there is a lack of discussion on carotenoids in algae.

22.   Line 242, “radiation includes UVA, UVB and UVC”. What about “visible radiation”?

23.   I don't think chapter 8 does a good job of illustrating the biomedical applications of bioactive substances in algae. And the description of the hazards of UVA, UVB, and UVC is repeated with the previous one.

24.   “Discussion” should focus on the shortcomings and prospects in this research field, and what is discussed here is more applicable to the “Introduction” and “Biomedical applications” parts.

25.   Improve all tables, including headings, content, notes and spelling.

26.   There are many language problems in this manuscript, it is recommended to review and improve throughout.

Author Response

1.  What is the difference between UV, UV-vis and VIS? The descriptions of these terms in the title, abstract, keywords, and text of the article are inconsistent.
Response: Changes have been made to clarify the definition of each electromagnetic spectrum range.
2. Line 19, “bioactive compounds”, “photoprotective agents”, “carotenoids” and “mycosporine-like amino acids (MAAs)” are inclusion relationships rather than juxtaposition relationships.
Response: Comments were attended.
3. At the beginning of the introduction, the description is inconsistent with the references. Please check to see if the same problem exists in the rest of the manuscript.
Response: Comments were attended.
4. The Introduction section does not introduce enough background
Response: Comments were attended.
5.      Lines 52-56, the sentence is too long, break it down into 2 sentences.
Response: Changes have been made.
6.      Line 64, it seems that there is a pronoun problem here. Please change “which” to “those”.
Response: Thanks for the observation, changes have been made in order to correct the sentence.
7.      Figure 1, the wavelength range does not match the description in the text.
Response: Comments was attended.
8.      The subject is missing in many statements in the text, such as line 95, line 103, line 147 and line 180, please check and correct them.
Response: Changes have been made.
9.      Line 101,change “Astaxanthinis” to “Astaxanthin is”.
Response: Thanks for the observation, corrected.
10.   At the end of chapters 3, 4 and 5, replace “UV radiation” with “UVA radiation”, “UVB radiation” and “UVC radiation” respectively.
Response: Thanks for the observation, corrected.
11.   Line 121-128, the same sentence appears twice.
Response: Sentence deleted.
12.   Line 148, a space is required between the number “37-150” and the unit “Kj/m2”.
Response: Thanks for the observation, corrected.
13.   Line 150, the pronoun “it” is missing after “support”.
Response: Changes have been made.
14.   Table 1, are ABTS and BBM the extraction methods? And what’s the full name of BBM?
Response: Thanks for the observation, corrected.
15.   Line 177, remove “a” before “high protein content”.
Response: Thanks for the observation, corrected.
16.   In the section 7, the discussion on the species and activity of UV protective metabolites in algae is not complete.
Response: The section was corrected.
17.   Line 185, a space is required between “been” and “shown”.
Response: Thanks for the observation, corrected.
18.   Line 189-196, authors should avoid use of “we, our, us etc”.
Response: Changes have been made.
19.   Line 202-207, check and improve the statement.
Response: Thanks for the observation, section was improved
20.   What does Table 5 want to show? It seems to be inconsistent with the text.
Response: The text was correct and Table 5 was not included in the updated manuscript.
21.   In section 7.2, there is a lack of discussion on carotenoids in algae.
Response: Comments were attended.
22.   Line 242, “radiation includes UVA, UVB and UVC”. What about “visible radiation”?
Response: Comments were attended. The paragraph was rewritten.
23.   I don't think chapter 8 does a good job of illustrating the biomedical applications of bioactive substances in algae. And the description of the hazards of UVA, UVB, and UVC is repeated with the previous one.
Response:  Section 8 was rewritten.
24.   “Discussion” should focus on the shortcomings and prospects in this research field, and what is discussed here is more applicable to the “Introduction” and “Biomedical applications” parts.
Response: Comments were attended. 
25.   Improve all tables, including headings, content, notes and spelling.
Response: Comments were attended. 
26.   There are many language problems in this manuscript, it is recommended to review and improve throughout.
Response: Comments were attended. several changes were conducted in the manuscript to improve the content.

Round 2

Reviewer 1 Report

The paper has been well improved. But prior to publication, the language should be carefully polished again. In addition, the section 6 in line 174-233 could be further organized, as the UV-vis further consists of different light spectrum. 

Reviewer 2 Report

There are still some inconsistencies could authors verified:

line 144: ,,the effects of UVB irradiation on microalgae are shown, showing'' I propose to change: ,,the effects of UVB irradiation on microalgae are listed, showing ....

Table 1  please change the shortcut: ADN to DNA in al places.

Table 1 the last row: Synechocystis this is cyanobacteria genus - verify and correct what there should be written.

Table2 row 2: UVB damage.... this is an effect of UVB not protection mechanism? or it should be as Protection against inhibition of………?

lines 197-206 Spirulina is a commercial name of edible cyanobacteria of Arthospira genus. Please make such note in this paragraph.
They are filamentous not single cells, so the sentence is not precise according to your definition of microalgae. see the pictures. Pay attention that this is the leading photosynthetic microorganism commercially produced on the Earth

Line 201-202 please give a citation concerning the properties of this organism here I suggested one example. doi:10.3389/fmicb.2017.02541

Line 207 to this paragraph about monochromatic light: the summary of effects on Arthrospira cultures is here chapter: Led Light of Different Colors: doi: 10.3389/fmicb.2017.02541

Line 251-254 Escitonemina, this is Scitonemin could you verify the whole text?

Line 255: references 51 and 52 these publications are about astaxanthin there is nothing about scytonemin. Please verify the references if they are properly assigned.

Line 297: substances change to: factors

Table 4 there is an error in references: ref 79 corresponds ref 80 in references list,  reference 75 in the table corresponds probably reference 76 in the references list this must be checked in the whole text.

Line 312 reference 60 this is a reference 61 in references list.

I think that all references are shifted from some point.

Line 348 I propose change: ,,they'' to ,,the producers''

Line 315-316 please for a reference to that.

Line 401: ,,protective mechanisms of microalgae against......'' or
,,effects of produced by microalgae compounds against.....''

  1.  

Reviewer 3 Report

the manuscript could be accept in current form